# Mitigating Scour at Bridge Abutments: An Experimental Investigation of Waste Material as an Eco-Friendly Solution

**Nadir Murtaza [1], Zaka Ullah Khan [1], Khaled Mohamed Khedher [2,*], Rana Adnan Amir [1], Diyar Khan [3,*], Mohamed Abdelaziz Salem [4] and Saleh Alsulamy [5]**

1. Department of Civil Engineering, University of Engineering and Technology Taxila, Taxila 47050, Pakistan; nadir.murtaza@students.uettaxila.edu.pk (N.M.); zakaullah.khan@students.uettaxila.edu.pk (Z.U.K.); ranaadnan_56@yahoo.com (R.A.A.)
2. Department of Civil Engineering, College of Engineering, King Khalid University, Abha 61421, Saudi Arabia
3. Department of Road Transport, Faculty of Transport and Aviation Engineering, Silesian University of Technology, 40-019 Katowice, Poland
4. Department of Mechanical Engineering, College of Engineering, King Khalid University, Abha 61421, Saudi Arabia; moabdulaziz@kku.edu.sa
5. Department of Architecture & Planning, College of Engineering, King Khalid University, Abha 61421, Saudi Arabia; s.alsulamy@kku.edu.sa
* Correspondence: kkhedher@kku.edu.sa (K.M.K.); diyar.khan@polsl.pl (D.K.); Tel.: +966-54-366-3078 (K.M.K.); Fax: +966-17-242-8184 (K.M.K.)

**Abstract:** Scouring around bridge abutments is a crucial and complex process that sometimes may lead to the failure of the bridge abutment. Therefore, in the present research, scouring around bridge abutments under clear water conditions was examined without and with countermeasures for providing an economical solution. A total of forty-five experiments were performed under clear water conditions to find the maximum scour depth around the bridge abutment. Experiments were performed in two different phases. In the first phase, scour depth was investigated without any countermeasures. In the second phase, scour depth was investigated using marble and brick waste as a countermeasure. The results showed that the maximum scour depth around the bridge abutment (at a distance of 10 cm on the upstream side and 15 cm on the downstream side of the abutment) for the Froude's number of 0.22 was 0.137 m without any countermeasure. The scouring depth increased up to 40% with an increase in the Froude's number from 0.13 to 0.22. The maximum reduction of scour depth was observed to be 40% and 55% when brick and marble waste were used as a countermeasure, respectively, compared to without a countermeasure case. It was concluded that marble and brick waste not only reduced scour depth to a significant level but also provided an economical solution.

**Keywords:** scouring; abutment; countermeasure; waste material; marble; bricks

## 1. Introduction

Bridge abutments play a crucial role in maintaining transportation infrastructure, but they are vulnerable to scour-induced damage, necessitating effective countermeasures. Scouring is a phenomenon when sand is washed away from bridge abutments and piers because of the hydrodynamic force of flowing water [1]. The main three causes of the erosion phenomena are the flow of water upstream, a horseshoe-like pattern at the foundation, and the wake vortex formed within the depth of scouring [2]. The scour holes are produced as a result of redirected flows of water through bridge piers and abutments located throughout the waterway, as their occurrence minimizes the overall shape and width of the water's path [3–6]. At the upstream end of bridge abutments, a downward stream is produced, which greatly increases the concentrated loss of soil right next to the foundation [7–10]. Local scouring at bridge abutments is predominantly caused by down-flow generated across the upstream edge of the abutment, which causes the formation of circular vortices at the foundation [11,12].

Previously, different researchers examined the reduction of scours around an abutment using different countermeasures under different flow conditions and bed material specifications. Collars and hooked collars have previously been investigated to prevent scouring around bridge piers [13–15]. The impact of the collar on the abutment had already been investigated, where a collar with a width of 2.25 $L_a$ ($L_a$ is the length of a short abutment) was placed, leading to the collar on a vertical wall beneath bed level. It was found that the height of the collar on the abutment concerning the sand bed elevation is a key scour depth reduction [16]. Collar installation was shown to be beneficial in preventing immediate collisions between downstream flows and the foundation material used. The impacts of erosive material were channeled away beyond the abutment, and the scour depth was reduced in the region of the abutment [17,18].

According to [19], a collar is a structure that functions as a barrier to prevent water from flowing downward, to weaken the force of semicircular vortices, and to prevent sediment from being removed from a pier and its abutments. According to [20], collars surrounding bridge abutments operate as safeguards against damage by being varied in thickness and forming the removal of sediment. According to [21], the dimension of the collar may be adjusted until it reaches a particular level before it could cause the water flow to become extremely turbulent and cause significant erosion. The various collar orientations were studied and discovered to have insignificant and ambiguous impacts on the creation of downstream flows [22–24]. Additionally, it was determined that the collar's diameter when placed surrounding the abutment affects how effective it is for minimizing erosion [7,24–26]. The Coanda-Effect-Based Polymetallic-Nodule Collector was used for sediment erosion and found that a collector performs slower when exposed to the flow for a long duration, and the collector's forward velocity is inversely proportional to the bed-sediment erosion depth [27].

However, different researchers also investigated the role of submerged vanes for scour protection around bridge abutments [28]. A combination of the submerged vane and riprap was used to evaluate their performance in scour reduction (54% around the abutment) [28,29]. Evaluate the scour reduction around the abutment by the position and height of the slot in the abutment. Different researchers [30–34] examined the role of riprap in scour reduction around the abutment. Riprap materials surrounding the bridge abutment, gabions, cable-tied blocks, and concrete mats or bags are examples of safeguarding systems [30]. When riprap was applied to the collared pier's foundation and its abutment, [31] observed that the movement of sediment was significantly reduced. They concluded that the scour depth might be reduced by up to 60% by installing two collared piers or abutments in a straight line perpendicular to the path of the water. It is important to put a solid filtration behind the surface that constitutes this riprap [32]. The water flow rate, the level of water, the riprap volume, and other factors all affect the dimensions of the boulders [33]. The absence of filtering, an insufficient gradient, a poorly designed bottom, the application of boulders of inappropriate dimensions, and other factors all contribute to the breakdown of the riprap approach [34]. Table 1 Summarizes the comparison of current research with previous research using traditional solutions for scour mitigation.

Therefore, the previous literature showed that traditional scour reduction techniques, including collars, riprap, and geo-bags, have been employed; however, these approaches can be cost-intensive, posing challenges for widespread implementation, especially in resource-constrained regions. As a result, current research focuses on the use of industrial waste materials, notably marble and brick waste, as cost-effective scour prevention strategies near bridge abutments are very important. The findings of this study could potentially change conventional scour reduction approaches in an era of sustainable infrastructure building, where economic and environmental considerations are key. The research provides a solution to improve bridge longevity while maximizing the utilization of resources by proving the cost-effectiveness and efficacy of these unusual resources compared to existing remedies.

**Table 1.** Comparison to Literature Review.

| Scour Countermeasures | Bridge Structure | Maximum % Age of Scour Reduction | References |
|---|---|---|---|
| Steel Collar and Geobags | Pier | 96 | [35] |
| Combination of collar and bed sill | Pier | 68 | [13] |
| Airfoil collar (Experimental) | Pier | 100 | [36] |
| Airfoil collar (Numerical) | Pier | 11–100 | [37] |
| Hooked collar | Lenticular Pier | 58 | [38] |
|  | Octagonal Pier | 73 | [14] |
| Spur dike | Abutment | 47 | [39] |
| Submerged vane | Abutment | 95 | [40] |
| Riprap and submerged vane | Abutment | 54 | [28] |
| Collar and slot | Abutment | 100 | [41] |
| Collar | Abutment | 77 | [23] |
| Collar | Abutment | 96 | [42] |
| Collar | Abutment | 87 | [43] |
| Collar | Abutment | 100 | [44] |
| Collar | Abutment | 100 | [45] |
| Collar | Abutment | 88 | [15,46] |

## 2. Material and Methods

### 2.1. Flume Characteristic

In the hydraulics laboratory of the Civil Engineering Department at the University of Engineering and Technology, Taxila, Pakistan, a laboratory study was conducted. The setup for the experiment was carried out in a channel that was transparent with conventional measurements of 1 m wide by 0.75 m deep by 20 m long, featuring transparent borders and a solid concrete bed, as indicated in Figure 1a. The underground tank's motor provided water to the channel. At the end of the flume, an inflated regular trapezoid sharp-crested weir was available for measuring discharge.

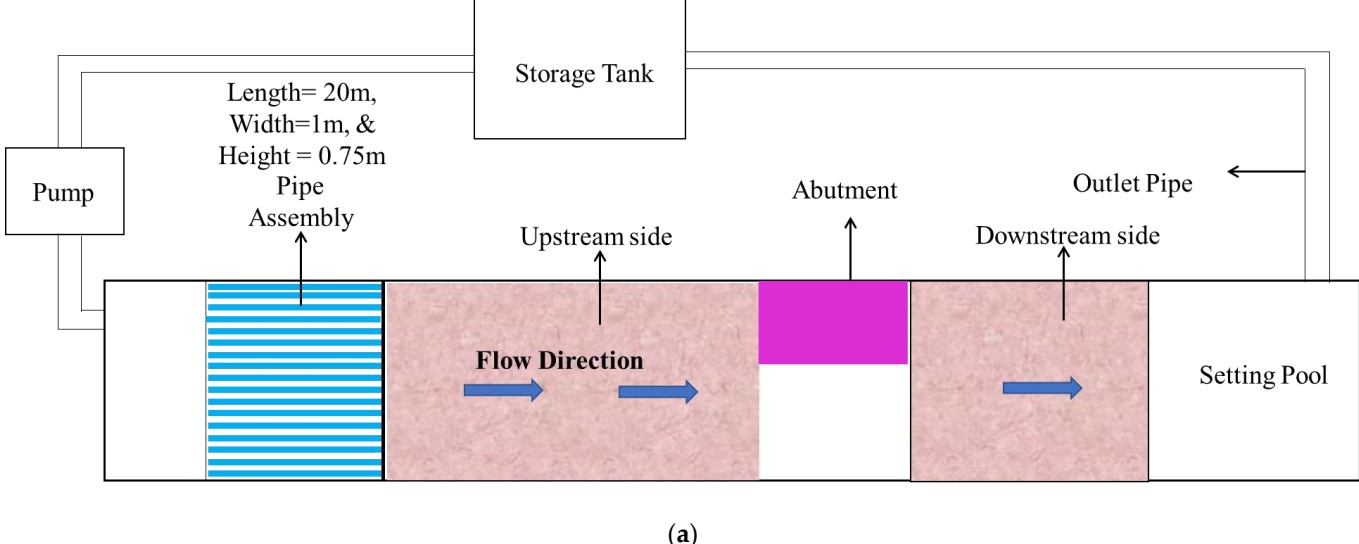

(**a**)

**Figure 1.** *Cont.*

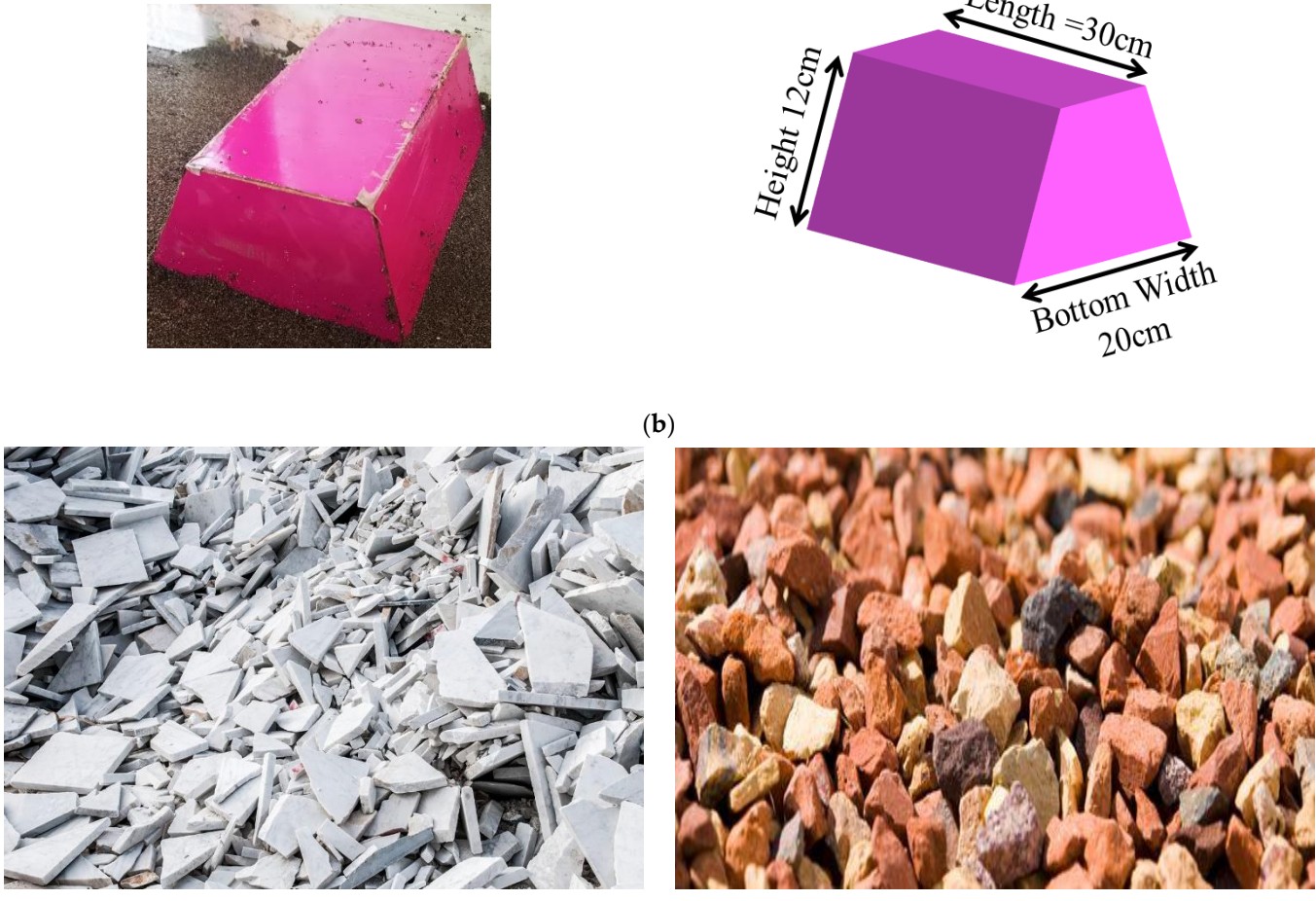

**Figure 1.** (**a**) Schematic diagram of the channel with the abutment model. (**b**) Abutment specification and schematic diagram. (**c**) Selected marble waste for the present research. (**d**) Selected brick waste for the present research.

### 2.2. Sediment Size and Flow Condition

The entire investigation was conducted on a homogeneous sand bed with a sediment size of 'd$_{50}$' of 0.51 mm with the help of a sieve analysis test. The geometrical average deviation of the grain size transportation, given by the equation g = (d$_{84}$/d$_{16}$)$^{0.5}$, was 1.24, where d$_{84}$ and d$_{16}$ are the particle sizes at 84% and 16%, respectively. The flow was allowed to keep bed stress from shearing below a certain level. The flow rate or discharge was measured with the help of a trapezoidal, sharp-crested weir fixed at the end of the channel. Five distinct discharges (0.019, 0.021, 0.023, 0.027, and 0.033 m$^3$/s) were used for the experiments. In each experimental examination, a flow level of 0.13 m was utilized across an abutment length of 0.30 m to meet the short abutment requirement, which is L$_a$/Y ≤ 1 [15]. For calculating the Froude's number, an average depth velocity was used, in which "V" is the flow velocity, "Y" is the water level in a flume, and F$_r$ = V/√gY was used for finding the Froude's number. The experimental condition for each test is shown in Table 2. Scour depth in each experimental run was measured with the help of a rail-mounted point gauge with an accuracy of ±0.1 mm. According to this requirement (U/U$_c$ < 1 [1]), the value of U/U$_c$ is 0.92 for the present investigation, wherein U is the approaching velocity of flowing and U$_c$ is the critical velocity for all tests that took place in

clear water. According to [1], the threshold velocity was estimated by using the velocity profile's log shape, and a similar procedure was adopted in previous studies [3,10,14,15].

$$\frac{U}{U_c} = 5.75 \times \log\left(5.53 \times \frac{Y}{d_{50}}\right) \tag{1}$$

where $d_{50}$ is the sediment size and Y is the depth of water in a flume.

**Table 2.** Experimental conditions and flow parameters.

| Q (m³/s) | $d_{50}$ | U (m/s) | Y (m) | $F_r$ | $U_c$ (m/s) | $L_a$ (m) |
|----------|----------|---------|-------|-------|-------------|-----------|
| 0.019 | 0.00051 | 0.14 | 0.13 | 0.13 | 0.008 | 0.3 |
| 0.021 | 0.00051 | 0.16 | 0.13 | 0.14 | 0.009 | 0.3 |
| 0.023 | 0.00051 | 0.18 | 0.13 | 0.16 | 0.010 | 0.3 |
| 0.027 | 0.00051 | 0.21 | 0.13 | 0.18 | 0.012 | 0.3 |
| 0.033 | 0.00051 | 0.25 | 0.13 | 0.22 | 0.014 | 0.3 |

*2.3. Abutment Specification*

The main component of the bridge structure is the abutment of the bridge, which is directly contrived by the action of scour throughout the normal and flash flood seasons. Due to this reason, abutment modification is especially essential in the present work, the most practical modification available in the literature. For the geometric similarities of the abutment model used in experimental work, the abutment located on the Kabul River has been taken as a reference. The Pushtoon Garhi Bridge Wing-wall Abutment on the Kabul River has a length of 2.6 m, a width of 1.3 m, and a height of 4.5 m [3]. The size of the Pushtoon Garhi Bridge Wing-wall Abutment on Kabul River was scaled down to 1/100 for selecting a wooden prototype for experimental work with a length of 30 cm, a top width of 10 cm, a bottom width of 20 cm, and a height of 12 cm, respectively. The details of the abutment are shown in Figure 1b.

*2.4. Marble and Brick Condition*

Several nations have significant worries about the use of uncontrolled landfill waste. As a result, multiple investigations are being carried out to discover a practical and ecologically sound remedy for waste that is solid. Recently, there has been a lot of interest in the use of recyclables as building materials. The qualities of clay-based bricks have been improved by the use of various wastes, including grain husk and biomass ash, ash from fly ashes, garbage waste, optical factory garbage, marble waste products, tissue mud, metal slag, and arsenic waste. The average size of marble available in Pakistan is 23 cm in length, with a thickness of 2–4 cm and a width of 10 cm [47], and the normal size of masonry bricks in Pakistan is 23 cm in length, 11 cm in width, and 7 cm in thickness, respectively [48]. In the current investigation, considering the above limitations, a marble waste of size 5 cm (Figure 1c) and a brick waste of size 3.5 cm (Figure 1d) were selected at a scale of 1/100. The marble and brick waste were placed at the upstream face of the abutment to counter the scouring depth surrounding the bridge abutment (Figure 2).

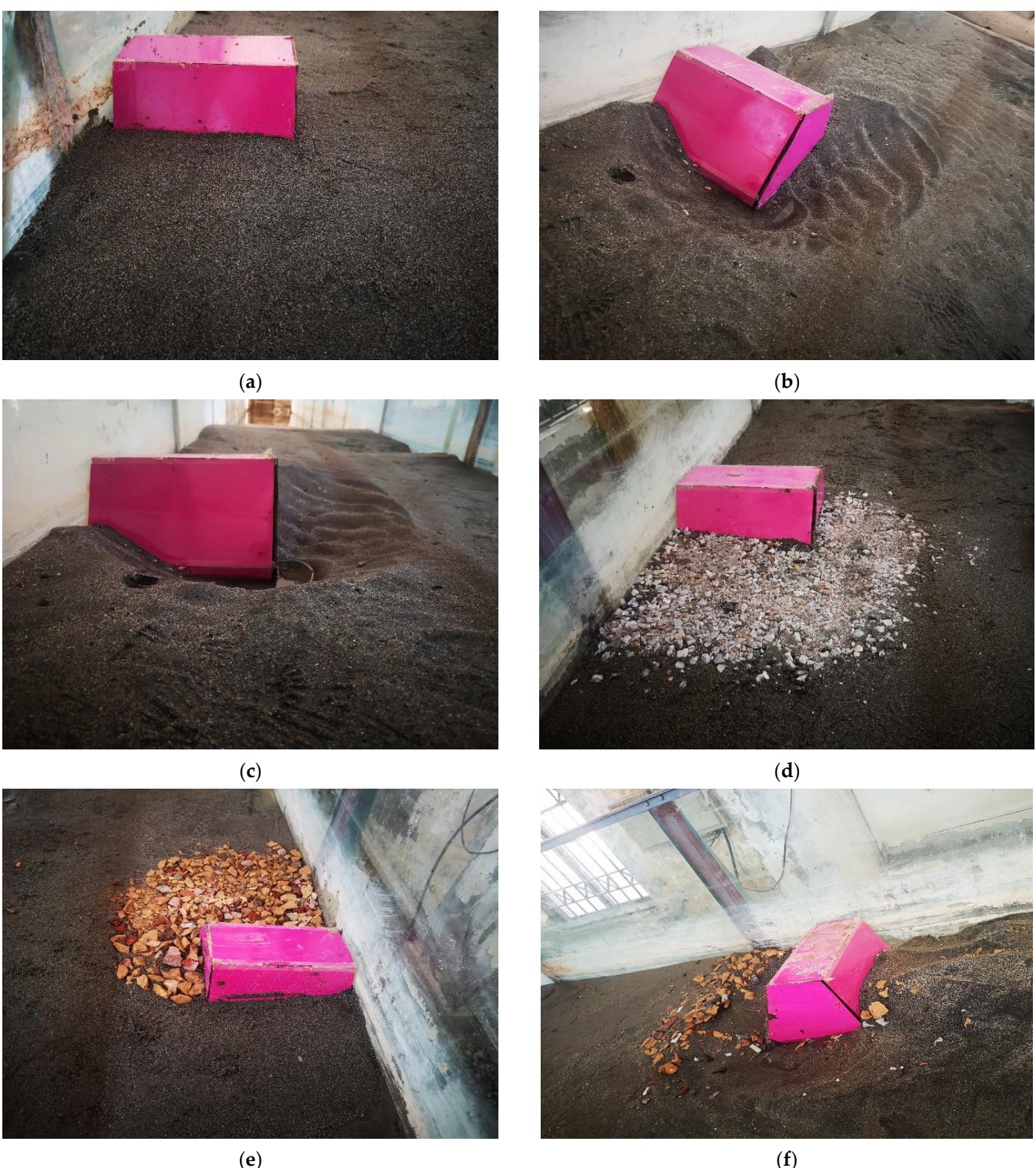

**Figure 2.** Laboratory setup of abutment model (**a**) Model setup in a flume (**b**,**c**) scour pattern around abutment (**d**) abutment model with marble waste (**e**) abutment model with brick waste (**f**) scour profile around abutment.

### 2.5. Laboratory Work Procedure

Each of the research activities was conducted under clear water conditions. To validate the characteristics of completely formed velocities, an abutment was placed at a distance of 7 m from the inlet of the experimental portion's sand layer zone [49]. A trapezoidal, sharp-

crested weir was set up using the formula established by [50] to estimate the magnitude of flows at the channel's terminus.

$$Q_t = \frac{2}{3}C_{rd2}\sqrt{2g}\left(bh^{\frac{2}{3}}\right) + \frac{2}{3}C_{rd1}\sqrt{2g}(b)h^{\frac{3}{2}} + \frac{8}{15}C_{td}\sqrt{2g}\left(\tan\frac{\theta}{2}\right)h^{\frac{5}{2e}} \tag{2}$$

where $\theta$ = notch angle and b = length of the weir

$C_{rd}$ = discharge coefficient of the rectangular, sharp-crested weir,
$C_{td}$ = discharge coefficient for triangular, sharp-crested weir
g = gravitational acceleration; h = water head on the weir crest; he = effective head

For every single experiment, the level of the water flow "Y" was maintained constant at 13 cm. The process of recording runs for the experiment's time began when every run achieved the necessary flow and flow of water level over the sand surface. A point gauge with a minimum count of 0.1 mm installed on a balanced carriage was used for determining the size of the hole caused by scouring around abutments (Figure 2a–f). A total of 45 experiments were conducted on bridge abutments without protection and with protection of marble and brick covering. The sand layer was uniformly leveled and 30 cm deep. The experiments were carried out in three stages. In the first step, abutments were tested without any sort of defense. The examinations on abutments using marble waste were carried out in the follow-up stage. The final stage included testing on abutments with brick waste defense. All of the preceding investigations were carried out on the five distinct flows indicated. The optimal equilibrium duration of every single experiment run was 6 h [51]. When the water had drained in the flume, the level of scouring around the abutment was recorded.

*2.6. Dimensional Analysis*

The effective parameters, such as initial water depth, Froude number, and percentage of bed material, were the main factors affecting the scouring phenomenon around the bridge pier [49]. According to [51], the scour depth downstream of the sluice gate depends on the jet Froude number, bed material size, and tail water depth. The maximum scour depth around the bridge abutment without and with countermeasures under clear water conditions is a function of the following parameters:

$$f(x) = (Y, d_{50}, V, U_c, A, ds, \rho, d, B, L, L_a, T, g)$$

In the above function, Y; the water depth in a flume, $d_{50}$; sediment particle size, V; an average flow velocity in a flume, $U_c$; critical velocity of flow in a flume, A; the cross-sectional area of the flume, $d_s$; the scour depth around the bridge abutment without and with countermeasures, $\rho$ is the density of the water, d is the sediment bed depth, B is the width of the flume, L is the length of the flume, $L_a$ is the abutment length, T is the time noted for each experimental trial, and g is the gravitational acceleration.

Buckingham $\pi$-theorem and dimensional analysis were used for deriving an equation considering the above-mentioned parameters, which can be represented as below.

$$f(x) = \left(\frac{QT}{Y}, \frac{Y}{VT}, \frac{V}{U_c}, \frac{A}{Y^2}, \frac{L}{B}, \frac{d}{d_{50}}, \frac{L_a}{B}, \frac{d_s}{Y}, \frac{d_s}{L_a}, \frac{V}{\sqrt{gY}}\right)$$

In the above function parameters such as $\frac{L}{B}, \frac{d}{d_{50}}, \frac{L_a}{B}, \frac{d_s}{Y}, \frac{d_s}{L_a}, \frac{V}{U_c}, \frac{A}{Y^2}$ are considered constant because they were not changed during experiments. Thus, the final equation derived can be represented as Equation (3).

$$f(x) = \left(\frac{Y}{VT}, \frac{V}{\sqrt{gy}}\right)$$

where $F_r = \frac{V}{\sqrt{gy}}$, therefore the above equation can be written as:

$$f(x) = \left( F_r, \frac{Y}{VT} \right) \tag{3}$$

## 3. Results

### 3.1. Scour Depth around the Bridge Abutment without Countermeasures

The scour depth around the bridge abutment is shown in Figure 3a. The maximum scour depth around the bridge abutment was 0.137 m when the Froude's number reached 0.22. It was noticed that the scour depth around the bridge abutment increases with increasing the Froude's number from 0.13 to 0.22 (Figure 3a). When the Froude's number increased from 0.13 to 0.22, it was noticed that scour depth also enhanced up to 40%. It was noticed that without any countermeasure, the scour depth starts from the upstream face (the face of the abutment towards the inlet of the flume whenever the abutment was placed in a flume) and increases towards the outer edge of the abutment, and then the eroded material starts deposition on the downstream side of the abutment. Figure 3b shows the maximum scour depth against each Froude's number without any countermeasures. Table 3 also summarizes the scour depth around the bridge abutment without any countermeasures. It was noticed that without any countermeasure case, the bridge abutment was fully exposed to the flow, and when the Froude's number increases from 0.13 to 0.22, which increases the intensity of flow and water interaction with sand particles at greater velocity, it results in significant scouring around the bridge abutment. It was also noticed that without any countermeasure, the scour depth around the abutment increases concerning time initially, and then a reduction in scour depth was observed with a time duration of 4.5 h. The contour map of each experimental case without countermeasures is shown in Figure 4.

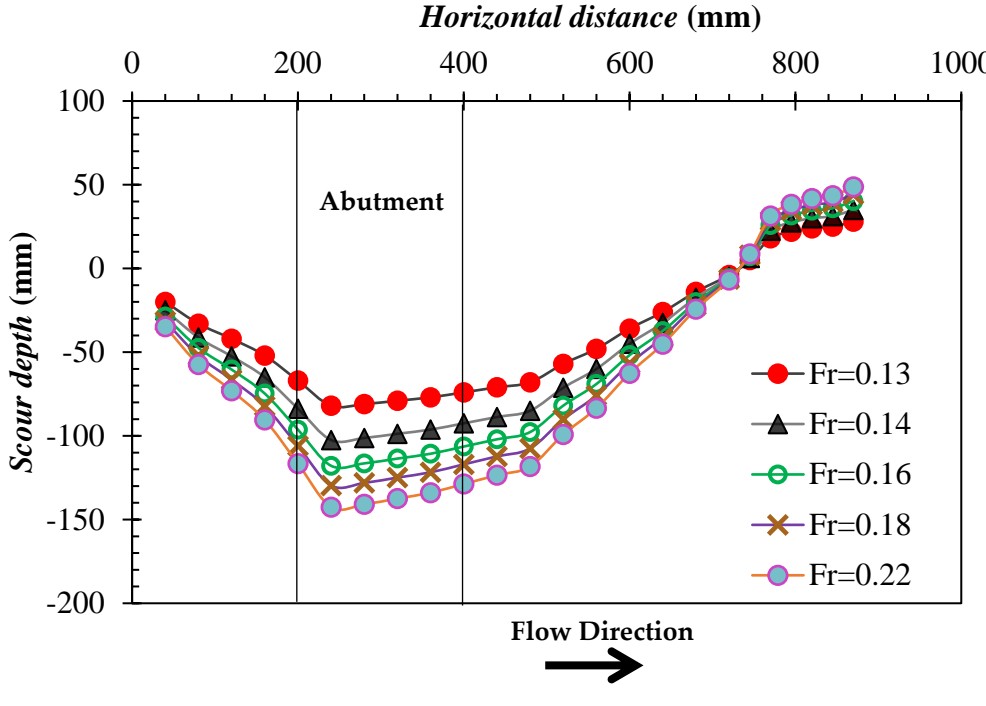

(**a**)

**Figure 3.** *Cont.*

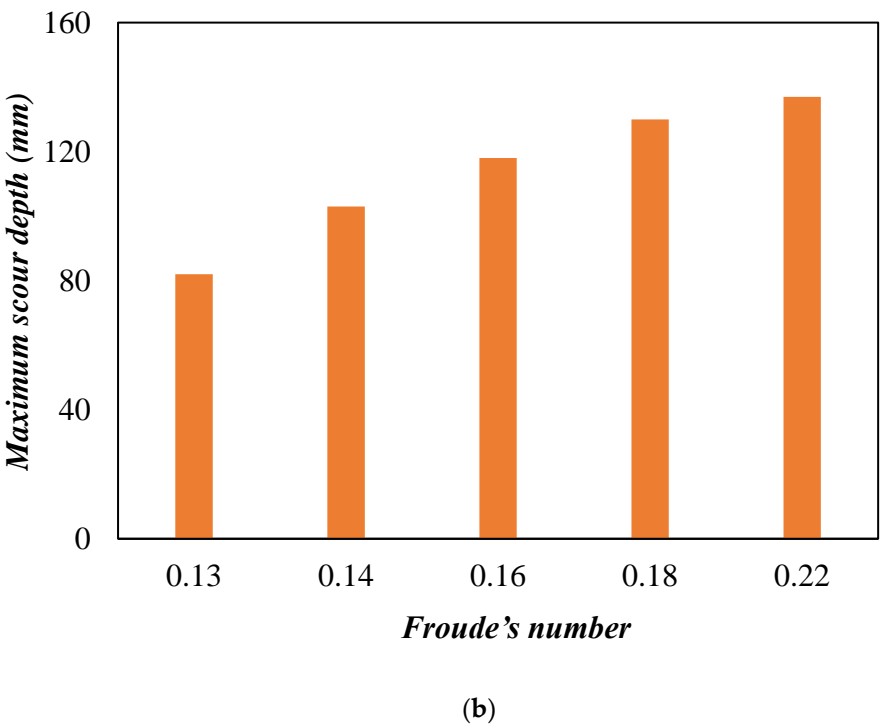

(**b**)

**Figure 3.** Scour around abutment without countermeasure with different Froude's numbers (**a**) scour profile around abutment (**b**) maximum scour depth around abutment without countermeasure.

**Table 3.** Results of different experimental cases.

| Froude's Number ($F_r$) | $d_s$ without Countermeasure (mm) | $d_s$ with Bricks Waste (mm) | $d_s$ with Marble Waste (mm) | $d_s/Y$ & $d_s/L_a$ (without Countermeasure) | $d_s/Y$ & $d_s/L_a$ (with Bricks Waste) | $d_s/Y$ & $d_s/L_a$ (with Marble Waste) |
|---|---|---|---|---|---|---|
| 0.13 | 82 | 49 | 37 | 6.3 & 2.7 | 3.7 & 1.6 | 2.85 & 1.23 |
| 0.14 | 103 | 62 | 46 | 7.8 & 3.4 | 4.7 & 2 | 3.56 & 1.53 |
| 0.16 | 118 | 71 | 53 | 9.06 & 3.9 | 5.5 & 2.4 | 4.08 & 1.77 |
| 0.18 | 130 | 78 | 58 | 9.9 & 4.32 | 6 & 2.6 | 4.49 & 1.94 |
| 0.22 | 137 | 82 | 61 | 10.5 & 4.55 | 6.3 & 2.7 | 4.71 & 2.3 |

*3.2. Scour Depth around Bridge Abutments with Marble Waste*

Figure 5a shows the scour depth around the bridge abutments when marble waste was used as a countermeasure. The maximum scour depth around the bridge abutment noticed was 0.061 m when the Froude's number was 0.22 (Figure 5a). It was noticed that using marble waste can reduce the scour depth up to 55%, and the scour depth increased with an increase in Froude's number from 0.13 to 0.22. The maximum and minimum scour depth were found to be 0.037 m and 0.062 m, respectively, using marble waste as a countermeasure. Compared to the case of abutment without any countermeasure, it was noticed that the scour depth reduced up to 55% when marble waste was used as a countermeasure (Figures 3a and 5a). This reduction was noticed because when marble waste placed on the upstream side of the abutment interacts with the flow, it starts flowing with water, but most of this waste is retained near the outer edge of the abutment and deposited, whereas some of the waste moves away from the abutment; hence, the retention materials protect the area around the abutment from scouring. Additionally, marble waste delays the scour process and scour hole development at the proximity of the abutment compared to without any countermeasure. Figure 5b shows the maximum scour depth around the bridge abutment for the range of the Froude's number from 0.13 to 0.22. The contour map of each experimental case when marble waste was used as a countermeasure

around the bridge abutment is shown in Figure 6. Table 3 also summarizes the scour depth around the bridge abutment using marble waste as a countermeasure.

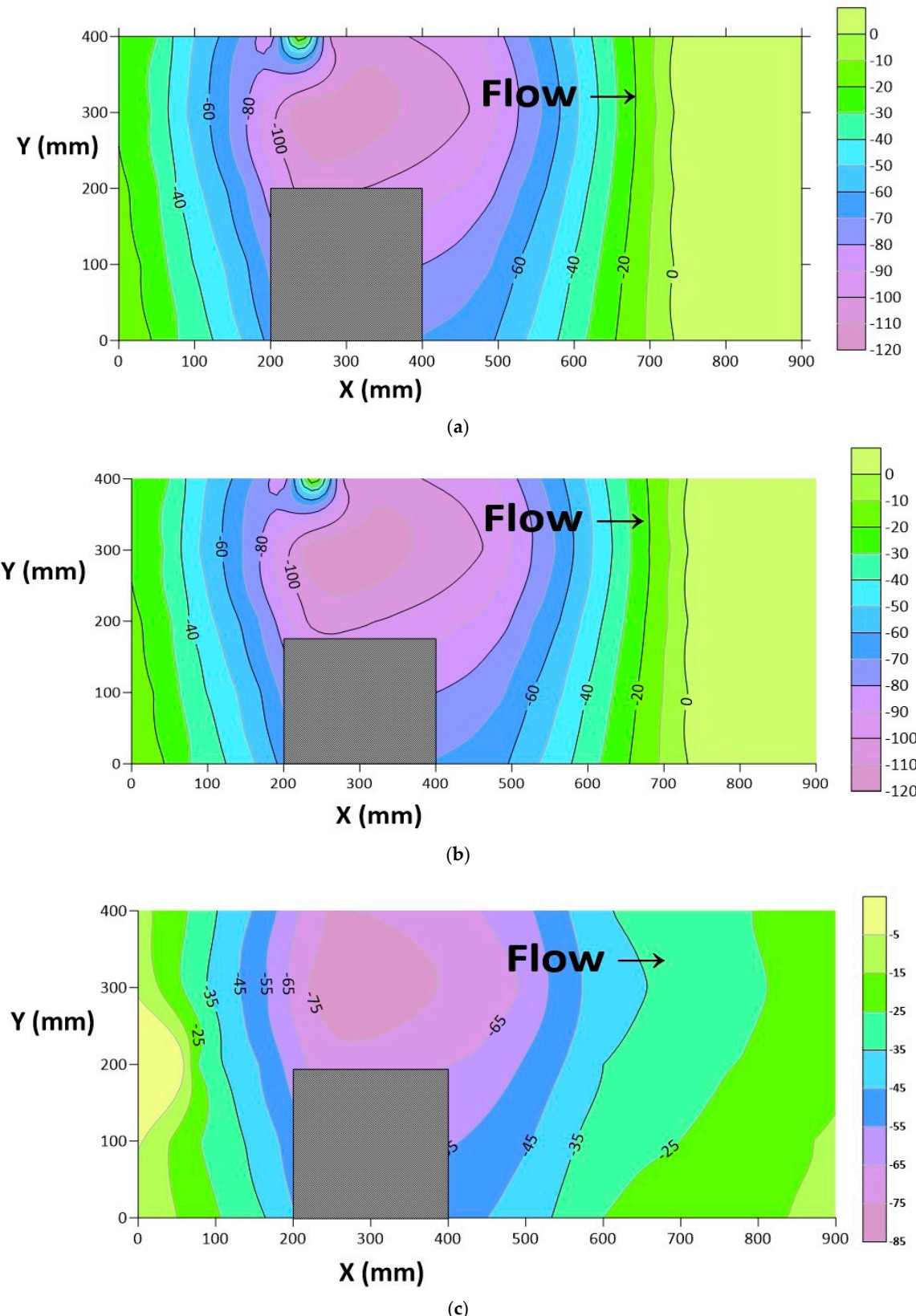

**Figure 4.** *Cont*.

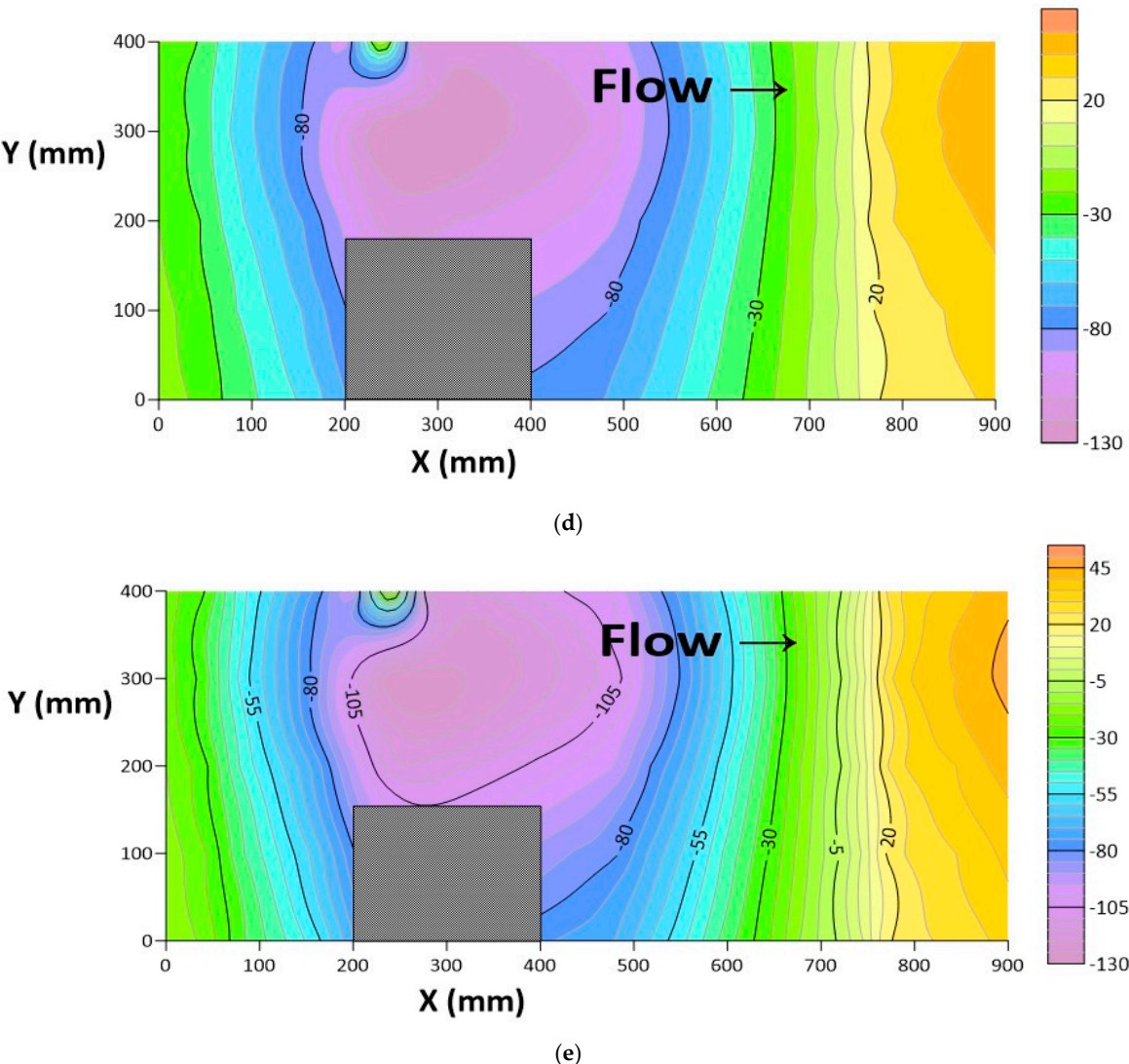

(**d**)

(**e**)

**Figure 4.** Contour of different experimental cases without countermeasure with different Froude's numbers (**a**) $F_r$ = 0.13 (**b**) $F_r$ = 0.14 (**c**) $F_r$ = 0.16 (**d**) $F_r$ = 0.18 (**e**) $F_r$ = 0.22.

*3.3. Scour Depth around the Bridge Abutment with Brick Waste*

Figure 7a shows the scour reduction around the bridge abutment using brick waste as a countermeasure. The maximum reduction in the scour depth around the bridge abutment was noticed to be 40% using brick waste as a countermeasure. It was noticed that, compared to without any countermeasure, the scour depth was reduced using brick waste as a countermeasure, but with increasing the Froude's number, the scour depth also increased (Figure 7a). The maximum scour depth noticed was 0.082 m, using brick waste as protection around the bridge abutments. The maximum and minimum scour depth for the given range of the Froude's number (0.13–0.22) were noticed to be 0.049 m and 0.082 m, respectively. It was also noticed that scour reduction using brick waste was less than using marble waste as protection around the bridge abutment. This was because of the size of the sample selected, meaning that brick waste was 3.5 cm and marble waste was 5 cm. As a result, most of the particles in the case of using brick waste flowed with water, and a small number of particles retained, whereas in the case of marble waste, a greater amount of particles retained around the bridge abutments, which reduced the scour process and scour hole development. Figure 7b indicates the maximum scour for the given range of the Froude's number, using brick waste as protection. The contour map for the given range of

the Froude's number is represented in Figure 8. Table 3 also summarizes the scour depth using brick waste as a countermeasure.

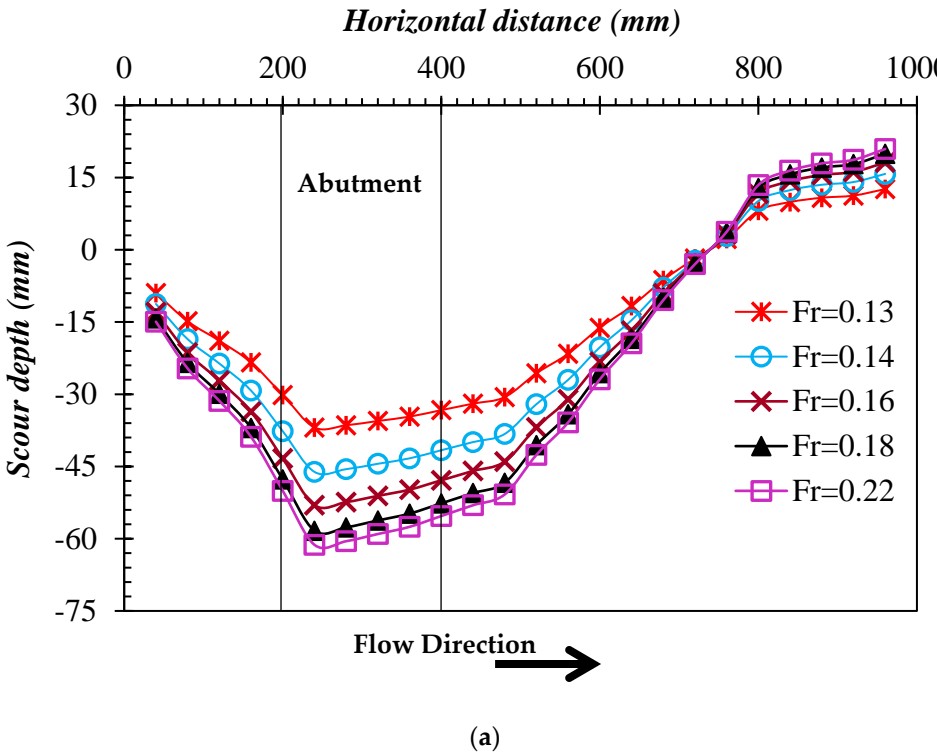

(**a**)

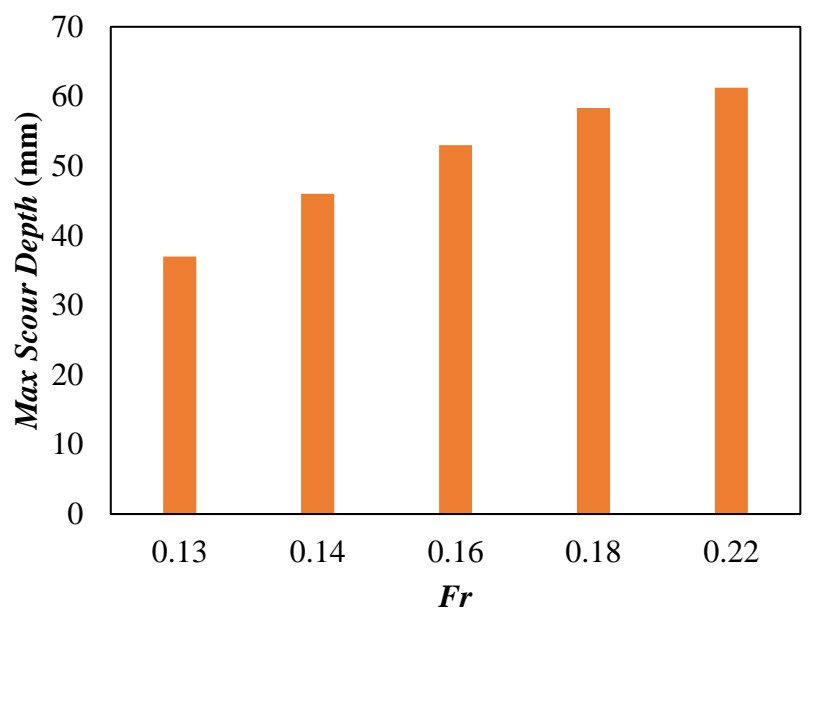

(**b**)

**Figure 5.** Scour around abutment under different flow conditions using marble waste as a counter-measure with different Froude's numbers (**a**) scour depth for different flow conditions (**b**) maximum scour depth around abutment.

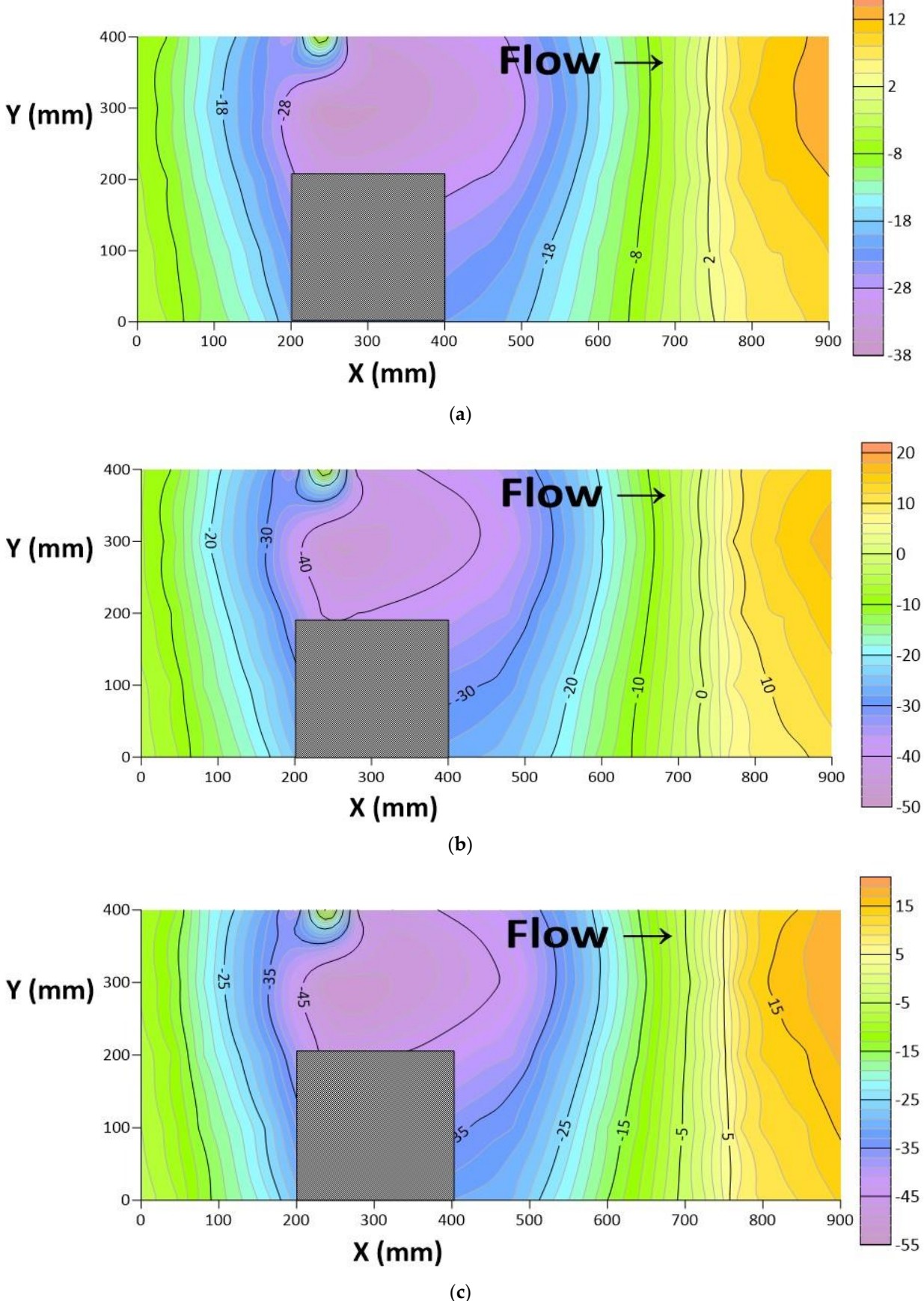

(**a**)

(**b**)

(**c**)

**Figure 6.** *Cont.*

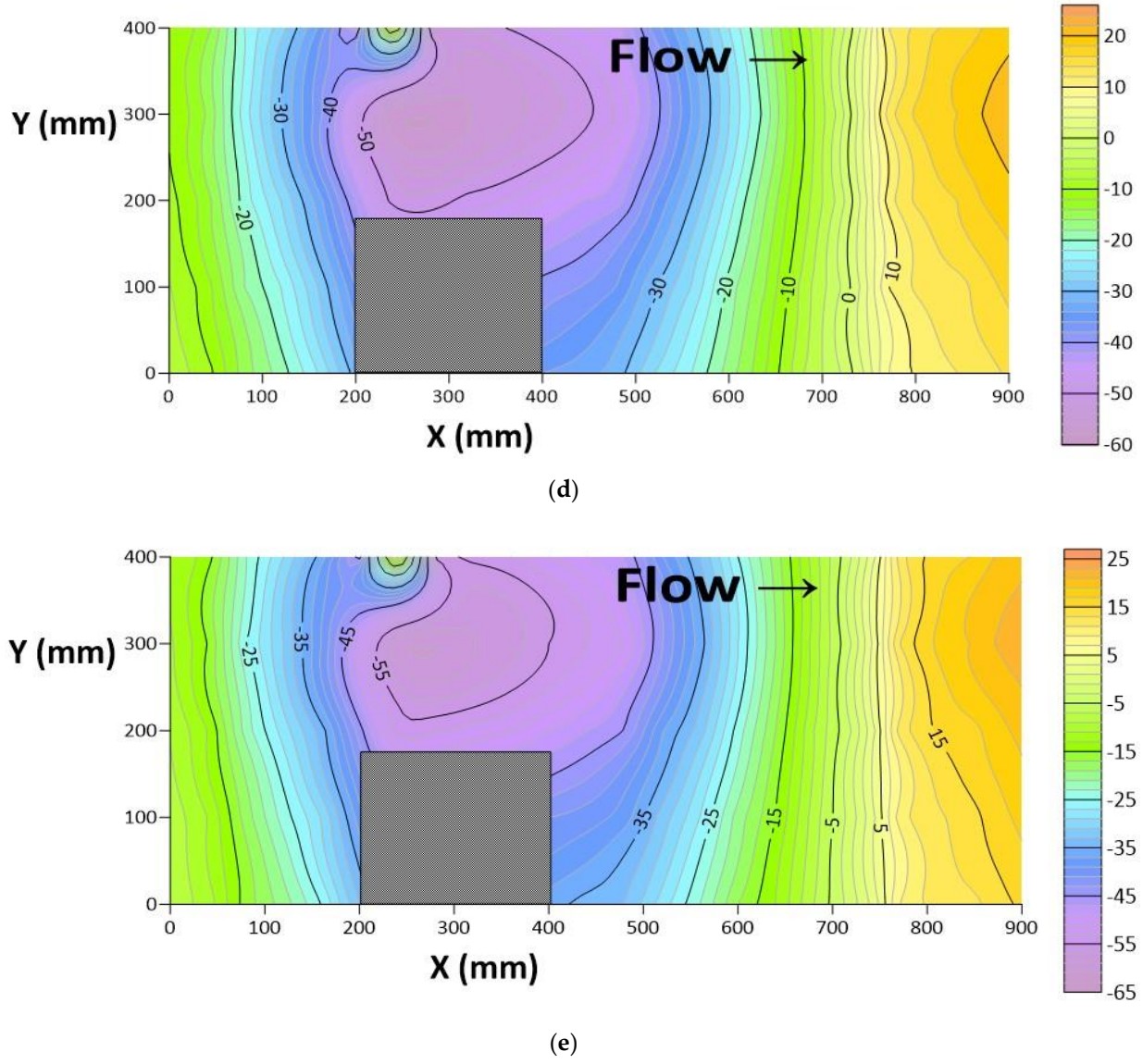

(**d**)

(**e**)

**Figure 6.** Contour of different experimental cases using marble waste as a countermeasure with different Froude's numbers (**a**) $F_r = 0.13$ (**b**) $F_r = 0.14$ (**c**) $F_r = 0.16$ (**d**) $F_r = 0.18$ (**e**) $F_r = 0.22$.

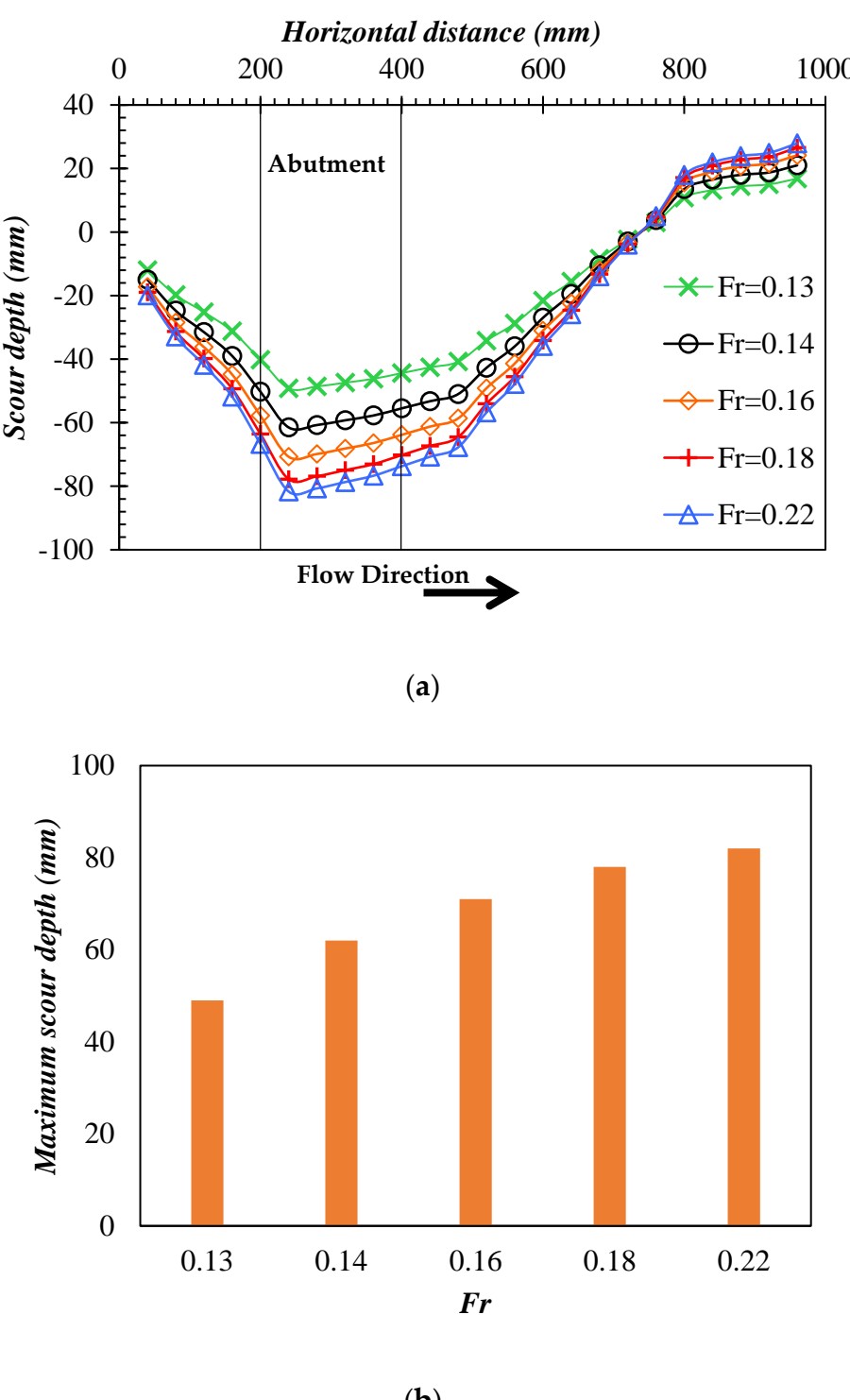

(**a**)

(**b**)

**Figure 7.** (**a**) Scour around abutment under different flow conditions with different Froude's numbers using brick waste as a countermeasure (**b**) maximum scour depth around abutment.

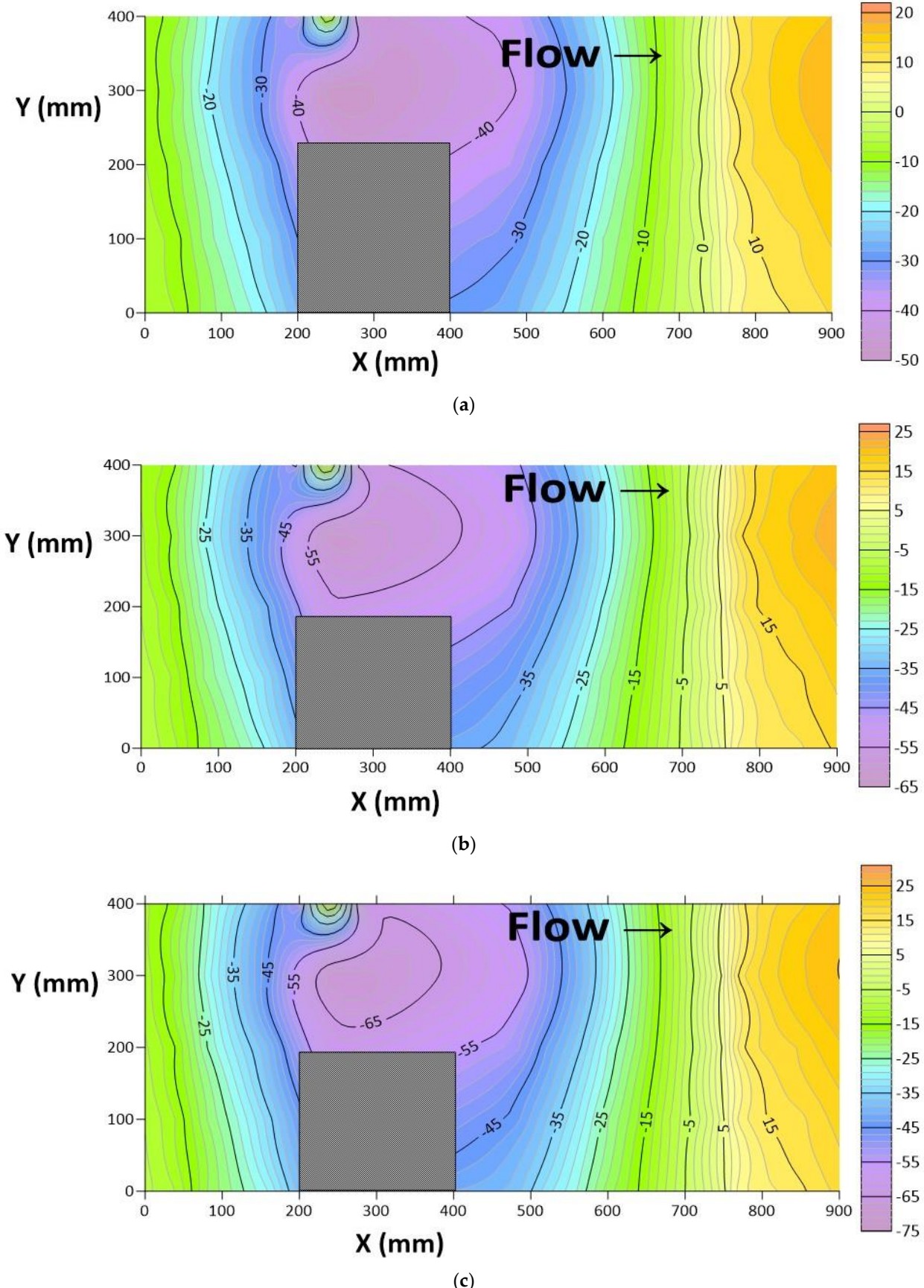

(**a**)

(**b**)

(**c**)

**Figure 8.** *Cont*.

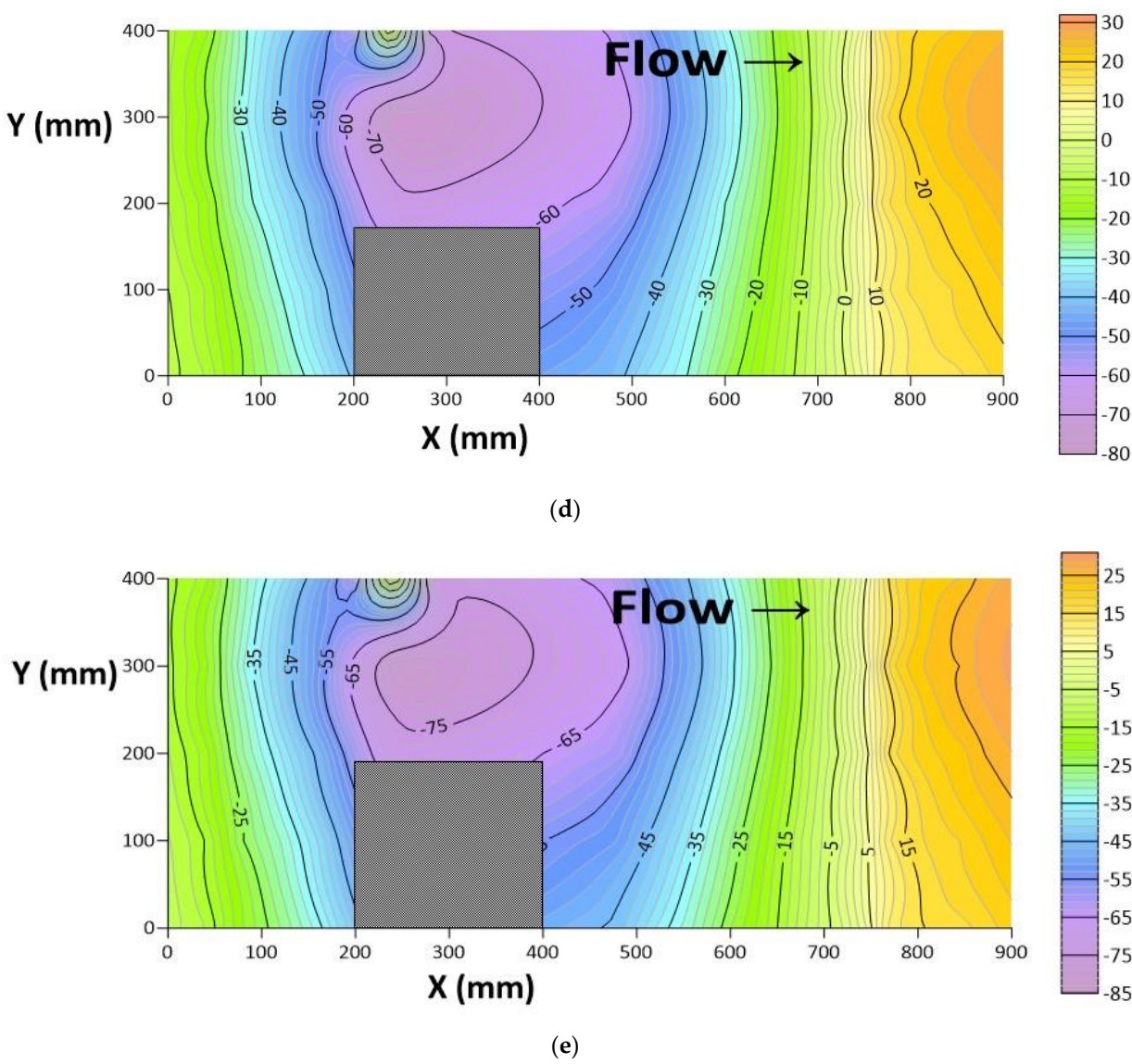

**Figure 8.** Contour of different experimental cases using brick waste as a countermeasure with different Froude's numbers (**a**) $F_r$ = 0.13 (**b**) $F_r$ = 0.14 (**c**) $F_r$ = 0.16 (**d**) $F_r$ = 0.18 (**e**) $F_r$ = 0.22.

### 3.4. Effect of Abutment Length and Water Depth on Scour

In the present study, the effect of abutment length and initial water depth has been investigated (Figure 9). The ratio of scour depth to abutment length ($d_s/L_a$) was observed to increase with increasing the initial Froude's number, and the ratio of scour depth to water depth ($d_s/Y$) increased with increasing the initial Froude's number (Figure 9). For considering without countermeasure, the maximum $d_s/L_a$ (4.55) and $d_s/Y$ (10.55) were observed for the Froude's number of 0.22 (Figure 9). The result showed that the length of the abutment along the transverse direction of the flume and water depth have a greater influence on scouring around the abutment. Similarly, the effect of abutment length and initial water depth has also been investigated with countermeasures (bricks and marble waste). It was observed that $d_s/L_a$ and $d_s/Y$ increase with increasing flow intensity (Froude's number). The maximum $d_s/L_a$ and $d_s/Y$ for Froude's number are 0.22 (Figure 9). The maximum $d_s/L_a$ and $d_s/Y$ for marble and brick waste were observed to be 2.30, 4.71, and 2.73, 6.3, respectively. The maximum reduction in $d_s/L_a$ and $d_s/Y$ when marble waste was used as a countermeasure was 55%, and for brick waste, it reduced up to 40% and 57%, respectively (Figure 9). Based on the initial Froude's number and water depth of flow, a Python code was run to generate Equations (4) and (5) for $d_s/L_a$ and $d_s/Y$ to

estimate the maximum scour depth around the abutment. To enhance the predictive power of the equations developed in this study, the valuable dataset, including the Froude's number and water depth of the previous research [52–54], was also entertained through Equations (4) and (5). For each dataset from previous literature, a standard deviation was determined to determine which dataset fit well with the reference line and the current study. According to Hosseini et al. [52], the dataset has a standard deviation of 0.84 and 0.67 for $d_s/L_a$ and $d_s/Y$, respectively. According to Osroush et al. [53], the dataset has a standard deviation of 0.75 and 0.57 for $d_s/L_a$ and $d_s/Y$, respectively. According to Saad et al. [54], the dataset has a standard deviation of 0.52 and 0.58 for $d_s/L_a$ and $d_s/Y$, respectively. For the current study, the dataset has a standard deviation of 0.25 and 0.40 for $d_s/L_a$ and $d_s/Y$, respectively. It was noticed from the comparison that the developed equations can predict $d_s/L_a$ and $d_s/Y$ significantly when required input parameters such as Froude's number and water depth are known. When compared to previous research, the comparison between studies not only confirms the accuracy of the findings but also highlights the creative methodology used in this study. Based on a predictive equation derived from Python code, predictive values of scour depth are shown in Figure 10.

$$\frac{d_s}{L_a} = 5.92 \times F_r{}^{0.8951} \times Y^{-0.3871} \tag{4}$$

$$\frac{d_s}{Y} = 6.6173 \times F_r{}^{0.5441} \times L_a{}^{-0.4148} \tag{5}$$

where, $d_s$, $L_a$, is the scour depth and length of abutment, and $F_r$, $Y$, is the Froude's number and water depth in a flume.

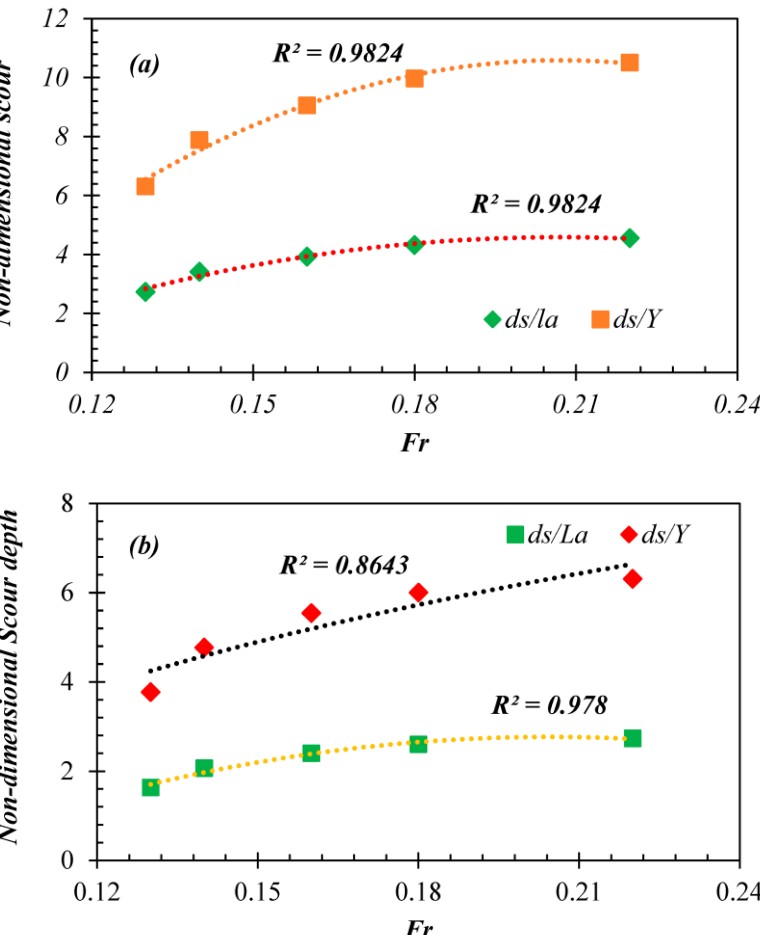

**Figure 9.** *Cont.*

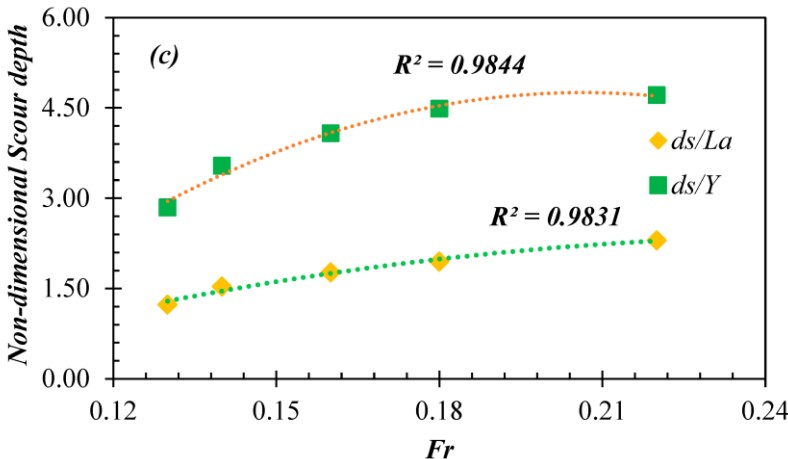

**Figure 9.** Effect of abutment length and water depth on scour depth around the abutment with different Froude's numbers (**a**) Without countermeasure (**b**) with brick waste (**c**) with marble waste.

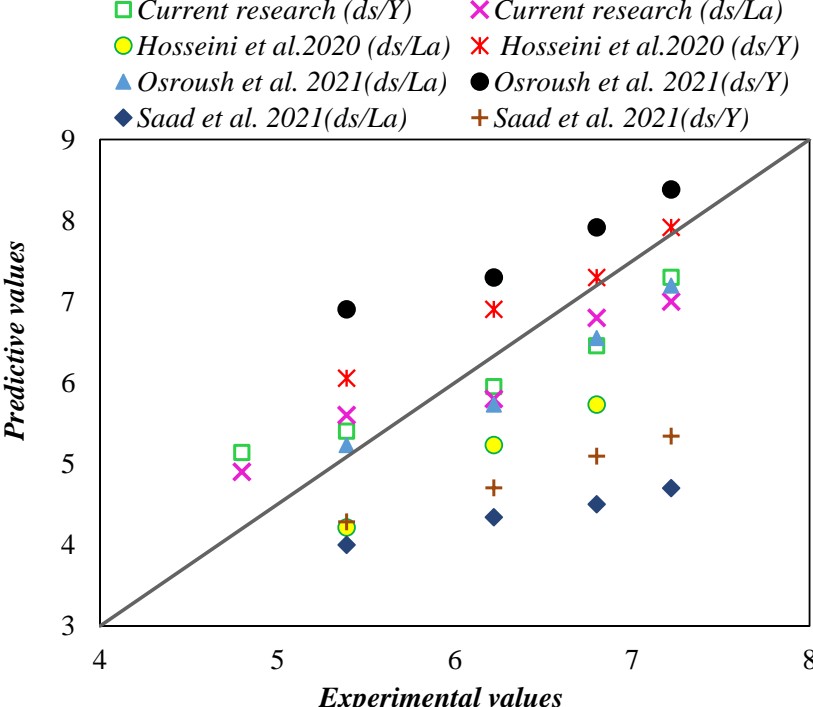

**Figure 10.** Predicated values of scour depth near the abutment calculated for [52–54] and current research using Equations (4) and (5).

### 3.5. Temporal Variation of Scour

Scouring around abutments is a time-dependent process. For the investigation of the temporal variation of scour depth around the abutment, an experiment was performed to examine the scour pattern for a period of 4.5 h (Figure 11). The results of each experimental case, that is, without any countermeasure and with countermeasures of marble and brick waste, showed that scour depth around the abutment increased after starting the flow in a channel, which gradually increased up to the maximum level and reached maximum scour depth after 4.5 h (Figure 9). The scour depths for different experimental cases with an interval of 30 min were observed and noted in a flume, as shown in Figure 11. The temporal evolution of the maximum scour depth of unprotected abutments and industrial waste material-protected abutments is presented in Figure 11. Initially, there is an increase in the scour hole of abutments until it reaches the maximum value. After that, a temporal

reduction in scour increment occurs. It is noticeable that 70–80% of maximum scour is attained within the first 2–3 h of the experiment. The performance of industrial waste material-protected abutments is much more satisfactory for scour reduction in comparison with abutments without protection.

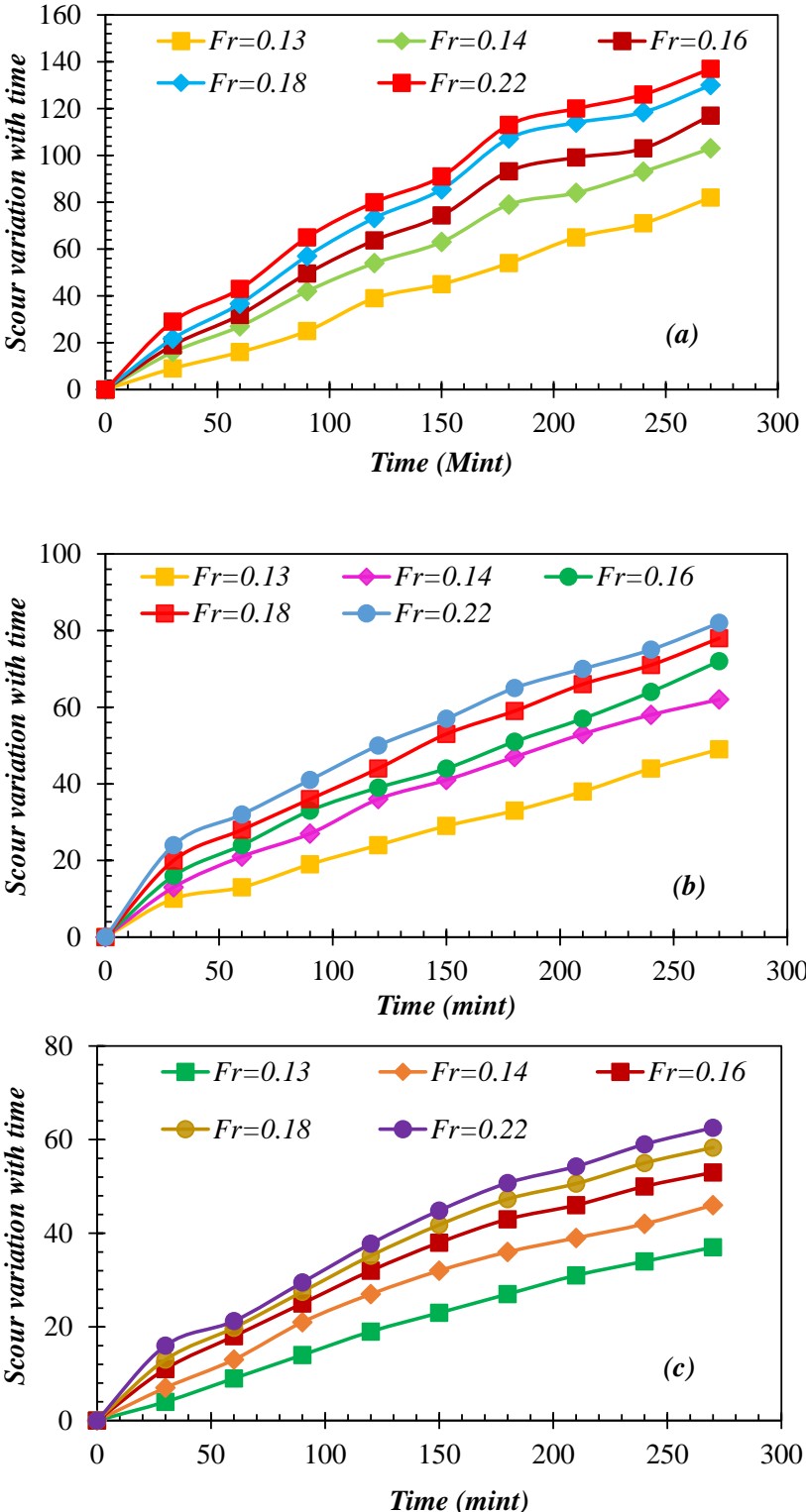

**Figure 11.** Temporal variation of scour around abutment with different Froude's numbers (**a**) without countermeasure (**b**) with brick waste (**c**) with marble.

## 4. Discussion

In the present study, scour reduction around the bridge abutment was investigated to provide a safe and economical solution using marble and brick waste as a countermeasure. The significance of this investigation was to examine the impact of marble and brick waste on scour reduction compared to already existing remedies in previous literature. In the past decades, different researchers have encountered scour reduction using different protections around the bridge to provide satisfactory solutions, such as [55–64]. The efficiency of scour defenses [57–59], pier-abutment interactions [60], the classification of the scour phenomenon in a compound flume [61,62], and abutment destruction [63–66] conducted extremely thorough laboratory studies on scour produced by both horizontal and upward flow limitation in compound channels. The preceding investigations concentrated on equilibrium scour instead of scour's temporal dependency. Investigations on the development of temporal scour are currently few and insufficiently thorough to make generic conclusions; significant research, notably [67,68], in particular, investigated temporal scour development at abutments in basic rectangular channels; the first study explored clear-water circumstances, while the latter analyzed live-bed conditions. [68] explored abutment scour development in compound channels; the former looked at thin-wall abutments, while the latter looked at more complicated abutment types (such as spill-through and wing-wall). It should be emphasized that [68] are the only researchers to look into scour development with interventions.

In the current research, the bridge abutment was initially tested without any protection around, and it was noticed that the scour depth around the bridge abutment continuously increased with increasing the Froude's number. Each experimental test was run for four and a half hours to examine the maximum scour depth around the bridge abutment. It was noticed that, when the bridge abutment was fully exposed to the flow without any protection, the maximum scour depth reached 0.137 m, which was initially lower and increased with duration, then started reducing and reached an equilibrium position after a time interval of four and a half hours. The percentage increment in the scour depth with increasing the Froude's number from 0.13 to 0.22 was noticed to be 40%, which reflects that with increasing the Froude's number, flow velocity increases and hits the sediment material around the bridge abutment with greater intensity, causing excessive erosion. To encounter the scour around the bridge abutment, first marble waste was tested, and it was noticed that the scour depth reduced significantly, up to 55%, and when brick waste was used as a protection, it was noticed that the scour depth reduced up to 40%, respectively. Using marble and brick waste as protection, the maximum and minimum scour depth around the bridge abutment were noticed to be 0.062 m and 0.037 m, 0.082 m, and 0.049 m, respectively. However, there was a 15% reduction in the case of marble waste compared to brick waste. The reason stated in the result was that marble waste used as a protection has a larger size (5 cm) and brick waste has a smaller size (3.5 cm) in comparison. Therefore, when water flowed across the abutment, it was noticed that some of the marble waste flowed with water and most of it retained around the bridge abutment, which resulted in a scour reduction, whereas in the case of the brick waste, due to its smaller size, most of the particles flowed with water towards the downstream side and some of the particles retained around the abutment, resulting in a smaller reduction. However, the size was limited for the present study; therefore, in the future, a larger size of brick waste should be tested to determine the efficiency and scour reduction percentage.

In order to provide a thorough description of the three-dimensional flow environment around a bridge foundation [69–71]. To forecast the flow pattern and scour near embedded hydraulic structures, extensive computational investigations have also been carried out. In a non-uniform gravel bed, ref. [72] investigated local scours using both experimental and observational data. They created a novel equation to predict the optimum depth of scour and recommended new K-factors for the Melville and Coleman relations. By calculating the overall sediment movement around a bridge pier exposed to local scour, ref. [72] developed a novel equilibrium scour depth relation. This strategy has been investigated by several

scholars, including [73–76]. The majority of this research sought to establish relationships for determining the pieces' sizes and location ranges. Thirdly, collars, slots, submerged vanes, and sacrificial piles are installed when the flow is changed or diverted from the foundation to lessen erosive forces [75–79]. These investigations sought to ascertain the geometric characteristics and efficacy of the suggested techniques. Some investigators have looked into combining the two approaches. According to [78], we evaluated the use of riprap individually and in conjunction with a collar as scour defenses around square bridge piers oriented with the flow. They created formulas for forecasting sustainable riprap thickness and width based on pier aspect ratios and offset angles. They determined that raising the inclined angle and aspect ratio of the piers increased the amount and quantity of riprap needed. Ref. [79] examined seven riprap thicknesses with a pair of distinct collar sizes for circular piers in identical trials (combination of riprap and collar). Their findings demonstrate that using a collar minimizes the size and extent of resilient riprap.

The current research also focuses on finding the impact of abutment length and water depth on the scour depth around the bridge abutment. It was noticed that scour depth increases with increasing the Froude's number in all cases, including without and with countermeasures around the bridge abutment. The ratios $d_s/L_a$ and $d_s/Y$ were determined in each case, such as without and with countermeasures. It was noticed that the ratios $d_s/L_a$ and $d_s/Y$ increased with increasing Froude's number, and the maximum values were 4.55 and 10.55, respectively. Similarly, $d_s/L_a$ and $d_s/Y$ were also determined using marble and brick waste, and the maximum reduction was noticed to be 55% and 57%, respectively. To examine the abutment length and water depth effects on the scour depth, an equation was developed. For this purpose, a Python code was generated considering three variables, including the Froude's number, water depth as the independent variable, and $d_s/L_a$ and $d_s/Y$ as the dependent variables. The equation was also used in previous research for comparison purposes [52–54]. It was noticed from the comparison that the developed equations can predict $d_s/L_a$ and $d_s/Y$ significantly when required input parameters such as Froude's number and water depth are known. When compared to previous research, the comparison between studies not only confirms the accuracy of the findings but also highlights the creative methodology used in this study.

## 5. Conclusions

In a laboratory investigation, using industrial waste materials like bricks and marble as a countermeasure for scour reduction around bridge abutments resulted in the following conclusion.

1.  The scour depth around the bridge abutment increased by increasing the initial Froude's number. The maximum increment in a scour depth without countermeasure for the range of Froude's number from 0.13 to 0.22 was observed to be 40%. This was because increasing Froude's number implies that an increase in a flow velocity that interacts with the exterior face of abutments directly results in significant scouring.

2.  When marble waste was used as a countermeasure, it was noticed that some particles (marble waste) flowed with water and some were deposited near the exterior face of an abutment, which caused resistance to the flow against scouring. Therefore, the maximum reduction in scouring around the abutment was noticed compared to without any countermeasure. The maximum reduction in scour depth was observed to be 55% when marble was used as a countermeasure. But scour depth increases with increasing flow discharge, and the minimum and maximum scour depths were 0.037 m and 0.062 m, respectively.

3.  The maximum reduction in scour depth was observed to be 40% when brick waste was used as a countermeasure. The scour depth increases with increasing flow discharge, and the minimum and maximum scour depths were 0.049 m and 0.083 m, respectively.

4.  In the case when marble waste was used as a countermeasure, it was noticed that scour reduction was 15% higher compared to the case when brick waste was used as a countermeasure. This difference in scour reduction was because of the sample size

in both scenarios. The size of the marble waste was larger than the brick waste, which was retained near the exterior face of the abutment, which caused a greater reduction in scour depth, whereas most of the brick waste used to flow with water and exposed the exterior face of the abutment after some time, hence causing a smaller reduction in scour depth.

5. For considering the case without countermeasure, the maximum $d_s/L_a$ (4.55) and $d_s/Y$ (10.55) were observed for the Froude's number of 0.22. The maximum $d_s/L_a$ and $d_s/Y$ for marble and brick waste were observed to be 2.30, 4.71, and 2.73, 6.3, respectively. The maximum reduction in $d_s/L_a$ and $d_s/Y$ when marble waste was used as a countermeasure was 55%, and for brick waste, it was reduced up to 40% and 57%, respectively.

**Author Contributions:** Conceptualization, N.M.; Methodology, N.M. and R.A.A.; Software, N.M.; Formal analysis, N.M. and R.A.A.; Resources, Z.U.K.; Data curation, Z.U.K.; Writing—original draft, N.M. and D.K.; Writing—review & editing, K.M.K. and D.K.; Visualization, D.K.; Funding acquisition, K.M.K., M.A.S. and S.A. All authors have read and agreed to the published version of the manuscript.

**Funding:** This research work was supported by the Deanship of Scientific Research at King Khalid University under grant number research group RGP. 2/463/44.

**Data Availability Statement:** Materials and data used in the present paper are available under request to the corresponding author.

**Acknowledgments:** The authors extend their appreciation to the Deanship of Scientific Research at King Khalid University for funding this work through a large scientific research group. The Project group number is RGP. 2/463/44.

**Conflicts of Interest:** The authors declare no conflict of interest.

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
