# Peer review of "Mitigating Scour at Bridge Abutments: An Experimental Investigation of Waste Material as an Eco-Friendly Solution"

_water, doi:10.3390/w15213798_

Round 1
Reviewer 1 Report
Comments and Suggestions for Authors
I found some serious shortcomings throughout the paper. The manuscript should be revised according to the attachment. The decision about the manuscript will be made after re-reviewing.

Comments on the Quality of English LanguageAuthor Response
Title: “Mitigating Scour at Bridge Abutments: An Experimental Investigation of Waste Material as Eco-Friendly Solution”.
Journal of Water (English Edition)
Respected Chief Editor,
Please find enclosed the revised manuscript entitled “Mitigating Scour at Bridge Abutments: An Experimental Investigation of Waste Material as Eco-Friendly Solution” that we would like to be considered for publication in Journal of Water (English Edition). Please find also a letter explaining, point-by-point, the changes made in response to the critiques/suggestions that we received.
We appreciate you and the Reviewers for your precious time in reviewing our manuscript and providing valuable comments. It was your valuable and insightful comments that led to possible improvements in the current version. We have carefully considered the comments and tried our best to address every one of them. We hope the manuscript after careful revisions meets your high standards. We welcome further constructive comments if any. Below we provide the point-by-point responses. All modifications in the manuscript have been highlighted in red.
We have made a concerted effort to adequately respond to each suggestion received from the Reviewers. We firmly believe that the Reviewer’s comments and suggestions have significantly improved this manuscript. We do hope that you and the Reviewers find this manuscript acceptable for publication in Journal of Water (English Edition).
Sincerely,
All Authors,

Reviewer 2 Report
Comments and Suggestions for Authors
This manuscript presents an experimental investigation on the reduction of local scouring around a bridge abutment using industrial by-product as a countermeasure. Local scouring of sediment around obstacles like abutments and piers is an important issue for hydraulic engineering and bridge engineering. Because of the complex flow structures including horse-shoe vortex, the local scouring is usually highly three-dimensional. The protection measures against local scouring are also very important. The experiments in this manuscript are well performed in general. However, the writing of the manuscript shall be substantially improved in my opinion. Detailed comments and suggestions are:
1 The “industrial by-product” in the title looks quite unclear and had better be replaced with more accurate words.
2 More detailed about the experiment shall be provided, such as the sediment particle size and the measurement methods and precision, and so on.
3 The non-dimensional derivation of scour depth has been well developed. The authors shall compare their results with published results and highlight the new contribute of this manuscript.
4 What’s the Reynolds number the experiments?
5 The plots in Figure 3 and 5, 8 can be combined.
Comments on the Quality of English LanguageThe English writing shall be improved throughout the manuscript.
Author Response
Please find enclosed the revised manuscript entitled “Mitigating Scour at Bridge Abutments: An Experimental Investigation of Waste Material as Eco-Friendly Solution” that we would like to be considered for publication in Journal of Water (English Edition). Please find also a letter explaining, point-by-point, the changes made in response to the critiques/suggestions that we received.
We appreciate you and the Reviewers for your precious time in reviewing our manuscript and providing valuable comments. It was your valuable and insightful comments that led to possible improvements in the current version. We have carefully considered the comments and tried our best to address every one of them. We hope the manuscript after careful revisions meets your high standards. We welcome further constructive comments if any. Below we provide the point-by-point responses. All modifications in the manuscript have been highlighted in red.
We have made a concerted effort to adequately respond to each suggestion received from the Reviewers. We firmly believe that the Reviewer’s comments and suggestions have significantly improved this manuscript. We do hope that you and the Reviewers find this manuscript acceptable for publication in Journal of Water (English Edition).
Sincerely,
All Authors,

Reviewer 3 Report
Comments and Suggestions for Authors
1. In abstract: “Scouring in the vicinity of the bridge abutment caused to affect the stability of the bridge abutments. To overcome this effect, in the current investigation, scouring in the vicinity of the bridge abutment in an open channel under different flow conditions was examined.” Please change it.
2. Revise the abstract thoroughly.
3. How are you going to define “Scouring in the vicinity of the bridge abutment”? How much area is going to be covered?
4. Symbols should appear according to the equations. Arbitrary written symbols and notations must be corrected.
5. Introduction section has no innovation and research organization.
6. They are a bit repetitive sentences in abstract and introduction and lack a smooth connection between them. Some paragraphs are too big; please separate them. There are many incorrect numberings in the subsection. Entire manuscript should be thoroughly revised. Correct the SI unit and correct format.
7. What inspired authors to use marble and brick waste (industrial by-products) as a countermeasure? One of the active research fields in countermeasure is collar please, mentioned in the literature (focus on latest literature such as airfoil and trapezoidal collar).
8. Figure should come where it is mentioned in the text.
9. Line 134: what is “acceptable flowing frequency (U/Uc)”?
10. Line 135: what is ‘optimum shear velocity U*c’?
11. Line 139 Where d50 is the sediment size and Y, is the depth of water in a flume. Correct it.
12. Section 2.6. Non-dimensional Pi-terms .. not clear and see equation 3
13. In Figure 3, the highlighted line represents the Froude number? Correct it in everywher.
14. What is meaning of legend? See below. Correct it everywhere.
15. Please revise the results and discussion section thoroughly. It becomes difficult to read and understand it.
16. Remove repetitive words from the conclusion section, which is already in the introduction
section.
17. Add your research findings to the Abstract section.
18. If possible, please change the title of the paper. It should be concise.
19. Some important and latest literature is missing from the manuscript. I have mentioned some recently published papers. Please refer below mentioned papers. These papers can be very useful for this review paper.
· Alhaddad, S., & Helmons, R. (2023). Sediment Erosion Generated by a Coandă-Effect-Based Polymetallic-Nodule Collector. Journal of Marine Science and Engineering, 11(2), 349.
· Gupta et al. (2023). Scour Reduction around Bridge Pier Using the Airfoil-Shaped Collar. Hydrology, 10(4), 77.
· Gazi et al. (2020). A review on hydrodynamics of horseshoe vortex at a vertical cylinder mounted on a flat bed and its implication to scour at a cylinder. Acta Geophysica, 68, 861-875.
· Gupta et al. (2023). Numerical simulation of local scour around the pier with and without airfoil collar (AFC) using FLOW-3D. Environmental Fluid Mechanics, 1-19.

Comments on the Quality of English LanguageMinor editing of English language required
Author Response

(The authors gave the same response as above.)

Round 2
Reviewer 1 Report
Comments and Suggestions for Authors
The manuscript is acceptable in present form.
Reviewer 2 Report
Comments and Suggestions for Authors
The authors have answered all my questions and made revisions in manuscript accordingly.
Comments on the Quality of English LanguageEnglish can be improved.
Reviewer 3 Report
Comments and Suggestions for Authors
The authors have made all the necessary changes. Hence MS can be accepted.
Comments on the Quality of English LanguageMinor English editing is needed.